# Fluorescent Polymers Conspectus

**DOI:** 10.3390/polym14061118

**Published:** 2022-03-11

**Authors:** Guillermo Ahumada, Magdalena Borkowska

**Affiliations:** Center for Soft and Living Matter, Institute for Basic Science (IBS), Ulsan 44919, Korea; m.borkowska000@gmail.com

**Keywords:** fluorescent polymers, fluorescent macromolecules, aggregation-induced emission, conjugated polymers, conducting electrolytes, conjugated polyelectrolytes

## Abstract

The development of luminescent materials is critical to humankind. The Nobel Prizes awarded in 2008 and 2010 for research on the development of green fluorescent proteins and super-resolved fluorescence imaging are proof of this (2014). Fluorescent probes, smart polymer machines, fluorescent chemosensors, fluorescence molecular thermometers, fluorescent imaging, drug delivery carriers, and other applications make fluorescent polymers (FPs) exciting materials. Two major branches can be distinguished in the field: (1) macromolecules with fluorophores in their structure and (2) aggregation-induced emission (AIE) FPs. In the first, the polymer (which may be conjugated) contains a fluorophore, conferring photoluminescent properties to the final material, offering tunable structures, robust mechanical properties, and low detection limits in sensing applications when compared to small-molecule or inorganic luminescent materials. In the latter, AIE FPs use a novel mode of fluorescence dependent on the aggregation state. AIE FP intra- and intermolecular interactions confer synergistic effects, improving their properties and performance over small molecules aggregation-induced, emission-based fluorescent materials (AIEgens). Despite their outstanding advantages (over classic polymers) of high emission efficiency, signal amplification, good processability, and multiple functionalization, AIE polymers have received less attention. This review examines some of the most significant advances in the broad field of FPs over the last six years, concluding with a general outlook and discussion of future challenges to promote advancements in these promising materials that can serve as a springboard for future innovation in the field.

## 1. Introduction

Luminescence is the emission of visible, ultraviolet, or infrared light in the optical range that is an excess over the thermal radiation emitted by the substance at a given temperature and continues after absorbing the excitation energy for a time significantly longer than the period of the absorbed light [1]. Various types of luminescence can be identified (e.g., hemi-, bio-, tribo-, and thermo-luminescence). Photoluminescence occurs when molecules interact with photons of electromagnetic radiation. Fluorescence occurs when electromagnetic energy is instantaneously released from the singlet state [2]. Some compounds display delayed fluorescence, which may be mistaken for phosphorescence. This is the outcome of two intersystem crossings, one from the singlet to the triplet and one from the triplet back to the singlet (Figure 1) [3].

The International Union of Pure and Applied Chemistry (IUPAC) defines fluorescence (for organic molecules) as the spontaneous emission of radiation (luminescence) from an excited molecular entity with spin multiplicity retention [4]. This definition becomes irrelevant in species such as nanocrystalline semiconductors (quantum dots or fluorescent quantum dots) [5] or metallic nanoparticles (fluorescent gold nanoparticles) [6] due to their complex emission processes.

Fluorescent materials have been in high demand over the last decade [8], and to meet this demand, a large number of substances with fluorescent properties have been explored, such as silica particles [9], glass [10], gold surfaces [11], quantum dots [12], and carbon dots [13], which are combined with a variety of chemical receptors to produce a variety of fluorescent materials. Because of this high demand, there has been a lot of interest in fluorescent polymers (FPs)research (Figure 2), due to their fascinating properties such as their increased signal response even after disturbance due to cooperative conformational effects of its chain segments. This is especially advantageous due to its superior visco-elastic and mechanical properties, which aid in the manufacturing of new devices [14,15,16,17,18,19]. FPs, similar to small fluorescent molecules, have a wide range of uses for sensing [20,21] and imaging [22,23,24,25,26,27], optoelectronics [28], fluorescent bioprobes [29], molecular imaging [30], photodynamic treatments [31], OLEDs [32], storage data security [33], encryption [34], anti-counterfeiting materials [35], and other fields [27,36,37,38,39,40,41,42,43].

Based on the strategies for developing these materials, two major branches can be distinguished in the field of FPs: (1) the use of a polymer chain (in general, using a conducting polymer) containing fluorophores [44,45,46,47,48], and (2) the burgeoning field of aggregation-induced emission (AIE) polymers [49,50,51,52,53,54]. Traditionally, conjugated polymers (CPs) have been employed as primary FPs [55,56], where the electronic conjugation between each repeating unit creates a semiconductive ‘‘molecular wire’’, providing very useful optical and electronic properties. However, the usage of non-conjugated polymers (NCPs) to create FPs has steadily gained popularity. This advancement was substantially aided by advances in controlled polymerizations of NCPs, which gave unprecedented control over polymer compositions and topologies [57,58].

In comparison with traditional FPs, AIE polymers present the advantages of high PL efficiency in aggregate and solid states, a large Stokes shift, outstanding photostability, etc. Thus, AIE polymers are expected to exhibit unique properties and remarkable advantages in their practical applications [51,59,60]. Furthermore, the structure, composition, and morphology of AIE polymers can be fine-tuned to meet the diverse needs of practical applications in chemo/biosensing, imaging, and theranostics.

Numerous reviews have been written independently for each type of FP, but classic FPs and AIE polymers are treated separately. This review discusses a variety of FPs in the hope that readers will gain a better understanding of the design strategy for FPs through a discussion of these papers. Finally, the challenges and future development of this class of materials are discussed.

## 2. Polymers Containing Fluorophores

### 2.1. Non-Conjugated Polymers Containing Fluorophores

Polymers are important and ubiquitous in modern society. They are widely used in housewares, packaging, coatings, biomedical supplies, textiles and fabrics, adhesives, engineering composites, and other applications due to their ease of processing and wide range of mechanical performances [61,62,63,64]. Polymers and polymer-based composites are designed and manufactured to be as robust as possible to meet the requirements of most engineering applications. Several examples in the literature can be found in the preparation of functional non-conjugated FPs by chemically customizing the core chromophore and the macromolecular assembly strategy.

Modulating photophysical properties through changes in environmental stimuli such as light, pH, pressure, heat, electric or magnetic fields, and chemical inputs is a growing area of research in FPs [65,66,67,68]. Mechanoresponsive luminescent (MRL) materials are interesting materials that change their emission color upon application of external forces. Weder and coworkers [69] introduced a novel approach that relies on an MRL compound combined with supramolecular polymerization. They proposed an alternative approach based on the derivatization of MRL chromophores with supramolecular binding motifs. The latter induces dyes to self-assemble into supramolecular polymers, which are then transformed into materials that combine MRL behavior with macromolecular mechanics. Cyano-substituted oligo(*p*-phenylenevinylene) (cyano-OPV)1,4-bis(α-cyano-4(12-hydroxydodecyloxy)styryl)-2,5-dimethoxybenzene was reacted with 2-(6-isocyanatohexylaminocarbonylamino)-6-methyl-4[1H]-pyrimidinone in hot pyridine, affording the supramolecular UPy-functionalized cyano-OPV, as shown in Figure 3a.

The material obtained showed the thermomechanical properties of a supramolecular polymer glass, emitting three distinct colors in solid state (red, yellow, and orange) with MRL and thermoresponsive properties (Figure 3b–f). It is hypothesized that the emission is influenced by molecular packing, which can be altered mechanically [70,71,72]. Controlling mechanochemical polymer scission with another external stimulus may provide a way to advance the fields of polymer chemistry.

Notably, light-driven reactions in conjunction with fluorescence-based techniques have become a significant synthetic tool in a range of chemical domains; changes in fluorescence are useful for monitoring reaction kinetics. Barner et al. [73] (Figure 4) developed a fluorescence-based methodology to analyze the kinetics of the step-growth polymerization in the photoinduced nitrile imine-mediated tetrazolene cycloaddition (NITEC). The tetrazole moiety rapidly interacts with activated dialkenes when exposed to UV light, resulting in a luminous pyrazoline-containing polymer. As a result, step-growth polymers’ fluorescence emission is proportional to the number of ligation sites in the polymer, resulting in a self-reporting optimal sensor system.

Figure 5 displays a conversion vs. reaction time plot obtained by ^1^H-NMR and fluorescence spectroscopy, indicating remarkable agreement between the two approaches. After 24 h, conversion rates of up to 90% were attained. In addition, the conversions for photopolymerizations in CDCl_3_ and THF-d_8_ exhibit extremely comparable tendencies.

This method is an exciting tool for monitoring the progress of a reaction, especially when NMR spectroscopy is challenging to use, such as when the backbone NMR resonances overlap with the resonances of interest, when the polymer’s solubility in common deuterated solvents is poor, or when high molecular weight polymers are analyzed.

Because of their good biocompatibility, high brightness, and ease of biofunctionalization, FPs have recently gained interest as imaging agents for biological applications; as a result, some examples of NCPs will be disclosed below. In 2015, X. Zhu et al. [74] prepared a set of multicolor fluorescent protein (GFP), by atom transfer radical polymerization (ATRP) [75] using an azide-modified polyethylene glycol macroinitiator (average molecular weight [Mn] = 12.3 kDa, polydispersity [Đ] = 1.21, yield = 57%). The free azide group was used to attach the fluorophores, following the well-known copper-catalyzed azide-alkyne-1,3- cycloaddition (click chemistry) [76].

The GFP with a color palette ranging from blue to orange was created using a combination of chemically tailoring the core chromophore, showing potential applications for fluorescent color regulation and cell imaging. GFP has received notoriety in biology as a genetically encoded noninvasive luminous marker [77] due to its minimal cytotoxicity and strong photostability. However, the macromolecular assembly showed the highest emission quantum yield (QY), approaching 8%, which is more than 80-fold greater than the core chromophore. The low QY values are attributed to the segmentation effect of polymers, which can diminish intermolecular contacts that quench fluorescence and hinder conformational free rotation (Figure 6).

Furthermore, developing effective drug-delivery vehicles is still a difficult task in materials science [78]. Serrano and coworkers [79] described poly(amidoamine) (PAMAM) dendritic core, functionalized with 2,2-Bis(hydroxymethyl)propionic acid (bisMPA) dendrons containing cholesterol and coumarin moieties, resulting in a new class of amphiphilic hybrid dendrimers. Their self-assembly activity was studied in both bulk and water. Because of their perfect macromolecular structure and precise amounts of functional groups, dendrimers are attractive candidates for medicinal applications [80] (Figure 7).

The synthesized dendrimers created spherical micelles in water due to their amphiphilic nature. The hydrophilic PAMAM cores are exposed at the surface, and the hydrophobic sections (coumarin and cholesterol moieties) remain inside the micelle. The cell survival of the micelles was examined in the HeLa (Henrietta Lacks) [81] cell line as a function of concentration, and all the micelles were shown to be non-toxic after 72 h of incubation at concentrations below 0.75 mg/mL.

In the same line, the group of Kanazawa [82] reported a reversible temperature-induced phase transition of N-isopropyl acrylamide (NIPAAm) copolymers with a fluorescent monomer based on fluorescein (FL), coumarin (COU), rhodamine (RH), or dansyl (DA) skeleton, employed as a molecular switch to regulate fluorescence intensity. Furthermore, pH responsiveness was seen in polymers with FL and COU groups, respectively (Figure 8).

The polymers were synthesized via radical polymerization in dimethylsulfoxide, using azobisisobutyronitrile (AIBN) and 3-mercaptopropionic acid as the radical initiator and chain-transfer agent, respectively. The weight-averaged molecular weight was determined by gel permeation chromatography (GPC) (Table 1).

The authors concluded that the switchable emission responses of these polymers were driven by a combination of the properties of both PNIPAAm and the fluorescent molecules. In addition, in vitro experiments into cultured macrophage cells (RAW 264.7) showed that intakes occurred over the lower critical solution temperature (LCST) over 30 °C for the polymers. This behavior is due to a significant increase in cellular absorption of FPs at this temperature, which appears to be caused by dehydration of polymer chains, as found in prior research by the group [83,84].

Additionally, following the success in intracellular thermometry, biologists realized the significance of temperature at the single-cell level and demanded that chemists produce more user-friendly fluorescence thermometers [23,85,86]. Recently, Uchiyama et al. [87] (Figure 9) have developed a cationic fluorescent nanogel thermometer (CFNT) based on thermo-responsive N-isopropylacrylamide and environment-sensitive benzothiadiazole (N-(2-{[7-(N,N-Dimethylaminosulfonyl)-2,1,3-benzothiadiazol-4-yl](methyl)amino}ethyl)-N-methylacrylamide; DBThD-AA) [88], exhibiting a great sensitivity to temperature fluctuations in live cells and a remarkable ability to penetrate live mammalian cells in a short incubation period (10 min).

This novel fluorescent nanogel thermometer is photobleached resistant in live cells, making it appropriate for long-term internal temperature monitoring, allowing intracellular temperature during cell division.

### 2.2. Conjugated Polymers Containing Fluorophores

Conjugated polymers’ capacity to act as electronic materials is based on the effective transport of excitons across the polymer chain [89]. In general, the excited state behavior of the corresponding conjugated polymers is dictated by the photophysics of the chromophore monomer. The influence of excited-state lifetimes and molecular conformations on energy transfer is investigated using various molecular architectures [90]. The opportunity lies in the nearly limitless possibilities for producing novel materials for specific uses by merely chemically adjusting the molecular structure. Conjugated polymers can achieve electrical qualities equivalent to non-crystalline inorganic semiconductors; nonetheless, conjugated polymers’ complicated chemical and structural features are nontrivial and mirror those of biomacromolecules. As a result, molecule conformation and interactions are critical to the operation of these material systems.

A conjugated carbon chain is a sequence of alternating single and double bonds, with highly delocalized, polarized, and electron-dense π bonds driving its electrical and optical activity. Polyacetylene (PA), polyaniline (PANI), polypyrrole (PPy), polythiophene (PTH), poly(para-phenylene) (PPP), poly(-phenylenevinylene) (PPV), and polyfuran are examples of common conducting polymers (CPs) (Figure 10). Synthetic conducting polymeric materials are widely used in a variety of applications, including packaging, adhesives and lubricants, microelectronic electrical insulators, and implanted biomedical devices [91]. Shirakawa et al. [92] discovered the first conducting polymer, a halogenated derivative of poly(acetylene), opened the door to a burgeoning area of fascinating new uses. The charge carrier mobility (which can be increased by doping) and significant light absorption in the UV–visible range are attributed to the delocalized (conjugated) electronic structure of poly(acetylene). Unfortunately, poly(acetylene) is difficult to manufacture and unstable in the presence of oxygen or water, making it unsuitable for a variety of applications. [93] Moreover, polyacetylene doped with bromine has a million times the conductivity of unadulterated polyacetylene, and this research was recognized with a Nobel Prize in 2000. CPs have recently been developed for use as roll-up displays for computers and mobile phones, flexible solar panels to power portable equipment, and organic light-emitting diodes in displays, in which television screens, luminous traffic, information signs, and light-emitting wallpaper in homes are expected to broaden the use of conjugated polymers as light-emitting polymers [16]. In this context, the insertion of inorganic atoms into the polymer chain is a powerful strategy for changing the characteristics of conjugated polymers. [94] In this context, Rupar et al. [95] reported the preparation of poly(9-borafluorene) vinylene (P9BFV, Figure 11) that showed simultaneous turn-off/turn-on fluorescence responses to fluoride in solution and NH_3_ (Figure 12) in the gas phase due to the presence of three-coordinate boron. It is important to emphasize that only a small number of conjugated polymers act as optical ammonia sensors [96,97,98].

The synthesis of the polymer P9BF was conducted via the Yamamoto reductive coupling [99] of Tip (Br). The polymerization is performed with small amounts of bromobenzene, which acts as a capping agent for the step-growth polymerization. On the other hand, the copolymer P9BFV was synthesized via Stille coupling [100].

Compared to the homopolymer P9BF, the prolonged conjugation of P9BFV, owing to the addition of the vinylene group, results in a lower optical bandgap (2.12 eV) and LUMO (4.0 eV, calculated by CV).

P9BFV has a strong solid-state fluorescence with an almost identical spectrum to that observed in the solution. The fluorescence changed from yellow to blue when the film was irradiated with UV light and exposed to ammonia. Furthermore, these alterations are reversible: removing the ammonia atmosphere within 5 min restores P9BFV’s fluorescence spectrum to its normal condition. When thin films of P9BF were exposed to ammonia, they behaved almost identically to P9BFV.

Along with CPs FPs materials, the Bielawski group reported the ring-opening metathesis polymerization of (bicyclo[2.2.2]octa-2,5,7-triene) (barrelene) to prepare copolymers with norbornene, producing robust films [101]. The copolymers prepared underwent spontaneous dehydrogenation in the presence of air or when exposed to laser pulses (phenylene vinylene), affording precisely defined fluorescent patterns with micrometer-sized dimensions. Direct laser writing (DLW) [102] on films of poly(barrelene-co-norbornene) produced a succession of well-defined patterns with micrometer dimensions, which were observed by the fluorescence of the conjugated polymer that formed after irradiation.

As demonstrated by various spectroscopic methods, PPV was produced by spontaneous dehydrogenation of the homo- and copolymers in the presence of air. According to a series of thermal studies, the copolymer displayed an exothermic response when heated to 100 °C, most likely due to oxidation (dehydrogenation), but did not undergo significant mass loss until around 400 °C (under a N2 atmosphere). Thermal aromatization was achieved in seconds after direct laser writing of the barrelene-containing copolymers by using a 400 nm diameter laser beam at a wavelength of 355 nm to write on the substrates that consisted of the copolymer at speeds of approximately 350 μm s^−1^. Raman spectroscopy was used to look at the dark and luminous portions of the copolymer to see if the laser aided in the oxidation process. The dark parts of Figure 13 showed signals consistent with poly(barrelene-co-norbornene). In contrast, the fluorescent sections showed signals compatible with PPV and consistent with the observations reported by the authors. This chemistry has the inherent advantage of allowing the monomer to be integrated into various macromolecular scaffolds and at different compositions, resulting in a diverse range of materials suitable for laser machining and current lithography applications.

Because of its increased detection sensitivity in the detection of a wide range of environmental contaminants and bioactive chemicals, conjugated polymer-based sensors have received a lot of interest [103,104,105,106]. Since the first conjugated polymer sensor was reported in the 1990s, this sensing platform has advanced more than two decades, with an explosion of research in this sector noted in the previous 10 years [107]. For instance, water-soluble conjugated polymers with ionic side chains are known as conjugated polyelectrolytes (CPEs) [108,109,110,111,112], and they have garnered attention in the past decade mainly because of their application as sensors, energy converters, and antimicrobials [113,114,115,116]. For example, Tan et al. [117] developed and synthesized four anionic CPEs with a poly(paraphenylene ethynylene) (PPE, Figure 14a) backbone but varied pendant ionic side chains. These CPEs were shown to attach to metal ions with varying selectivities via polymer–metal ion interactions, resulting in a wide range of fluorescence responses. Four CPEs were structurally and photophysically characterized and used as PPE sensor arrays. The sensor array’s fluorescence intensity responses were tested after the injection of eight different metal ions separately.

This approach generated environmentally friendly fluorescent materials, with easy sample preparation and measurements and convenient data processing with a simple pattern analysis procedure that distinguishes from other existing methods. The authors improved the methodology for detecting combinations of harmful metal ions, with the possibility of expanding to other metal ions in the future.

The development of new detecting methodologies based on FPs has also been extended to organic molecules. Tetracycline (Tc), for example, is a type of antibiotic that is commonly used in veterinary medicine, human treatment, and agriculture, and it is critical to measure in water. The development of simple and efficient methods for detecting and removing TC from water is highly desirable but continues to be a challenge. Because of major environmental concerns, including ecological hazards and human health consequences, tetracyclines are a special case among the many antibiotics used. Most data suggest that tetracycline antibiotics are ubiquitous substances found in many ecological compartments due to their widespread use [118,119]. More than 70% of tetracycline antibiotics are excreted and released inactive form into the environment after administration via urine and feces from people and animals. Because of their very hydrophilic nature and low volatility, they have demonstrated great endurance in the aquatic environment [120]. Iyer et al. [121] synthesized a CPE following a palladium-catalyzed Suzuki cross-coupling polymerization, poly[5,5′-(((2-phenyl-9H-fluorene-9,9-diyl)bis(hexane-6,1-diyl))bis(oxy))diisophthalate] sodium (PFPT), and used it for highly sensitive detection of tetracycline in water (Figure 15).

Due to the electron-rich nature of the polymer PFPT, cyclic voltammetry analysis revealed only the oxidation peak, from which the HOMO level was determined to be 5.95 eV. From the beginning of the UV–vis absorption spectrum, the optical band gap was calculated to be 3.35 eV, whereas the LUMO level was determined to be 2.6 eV. The HOMO (7.55 eV) and LUMO (4.53 eV) energies of tetracycline were also determined using cyclic voltammetry and onset UV–vis measurements. These results demonstrate that photoinduced electron transfer from the polymer’s LUMO (2.6 eV) to the quencher Tc’s LUMO (4.53 eV) is the sole mechanism by which Tc selectively quenches PFPT fluorescence.

In 100% aqueous media, the detection limit of PFPT toward Tc is reported to be 14.35 nM/6.80 ppb (6.8 ng/mL). The effective electron transport from PFPT to Tc via electrostatic/hydrogen bonding interactions resulted in a high quenching efficiency with a K_sv_ value of 1.57 × 10^5^ M^−1^.

As well as sensing materials [55], CPs have been intensively researched for biosensing and bioimaging applications due to their great light-harvesting capacity, outstanding photostability, and easily surface-modifiable features [122,123,124]. For instance, photodynamic therapy (PDT) [125,126] has become an important cancer treatment that uses reactive oxygen species (ROS) (such ^1^O_2_ and hydroxyl radicals) created by exposing photosensitizers (PS) to light irradiation in an oxygen-rich environment to destroy cancer cells. Zhang and coworkers [127] prepared a positively charged water-soluble polythiophene polymer (PT0, Figure 16). This exhibited high photo- and pH-stability, a sizeable two-photon absorption cross-section, and the capability to generate ^1^O_2_ (g).

Irradiation using 780 and 900 nm lasers of 406, and 473 mW of laser power, respectively, with prolonged irradiation (4 to 5 min) was needed to kill HeLa cells efficiently. Although PT0 does not appear to distinguish cancer cells from noncancerous cells, such selectivity might be achieved via a method such as combining PT0 with a cancer cell targeting probe, which will require additional investigation.

## 3. Aggregation-Induced Emission Macromolecules

Over the last two decades, there has been a lot of interest in aggregation-induced emission (AIE) [128,129]. Compared to the well-studied low-mass AIE luminogens, AIE polymers have received less attention, despite outstanding advantages such as high emission efficiency in aggregate and solid states, signal amplification, good signal amplification, good processability, and multiple applications, among others [51]. AIE polymers combine the benefits of polymeric materials with AIE luminogens, and they exhibit distinct features compared to standard fluorophores and low-mass AIE materials for practical applications. The primary technique for producing AIE polymers is to integrate aggregation-induced, emission-based fluorescent materials (AIEgens) [130,131,132] into polymer backbones or as polymer pendants via polymerization and modification.

Since the introduction of the AIE concept, coined in 2001 by Tang et al. [133], luminogenic materials with AIE properties have piqued the curiosity of many researchers. In principle, luminophores aggregation has two impacts on photoluminescence (PL): aggregation-caused quenching (ACQ) and AIE. Depending on the molecular architectures and intermolecular packing, the ACQ and AIE effects will compete in most luminogens (Figure 17). Generally, typical AIEgens, including tetraphenylethene (TPE), distryreneathracence (DSA), hexaphenylsilole (HPS), 1,8-naphthalimide, 2,4,6- triphenylpyridine, and tetraphenylpyrazine (TPP) are widely used as scaffolds to prepare AIE polymers.

Luminous liquid crystals (LLCs) combining intrinsic light-emitting properties and liquid crystalline ordering have piqued interest in recent decades due to their potential applications as emissive liquid crystal displays (LCDs), organic light-emitting diodes (OLEDs), sensors, and optical information storage. [135,136,137,138,139] Xie et al. [140], using AIE luminogens, were able to successfully manufacture a series of novel, highly efficient luminous liquid crystalline polymer (LCPs), namely poly{2,5-bis{[2-(4-oxytetraphenylethylene)-n-alkyl]oxycarbonyl}styrene}, with different chain lengths (m = 2, 4, 6, 8, 10, 12), as show in Figure 18. LCPs are polymers with liquid crystal properties and frequently contain aromatic rings as mesogens. Liquid crystallinity can also be found in polymeric materials such as LCEs (liquid crystal elastomers) and LCNs (liquid crystal networks). They are both crosslinked LCPs. However, their cross-link density is different [141].

Furthermore, the polymers demonstrated high-efficiency luminescence in the liquid crystalline state due to the “Jacketing” phenomenon, which strongly depended on the spacer length (solid-state quantum yields declined from 52 percent to 18 percent as spacer length increased). Meanwhile, as the length of spacers increased, the glass transition temperatures (Tg) decreased. The polymers had good film-forming and processing properties, making them promising materials for luminous devices. OLED technology might usher in a new age of large-area, transparent, flexible, and energy-efficient display and lighting goods.

Fluorescence-based techniques not only provide answers for the creation of novel materials, but also have potential biological applications due to their high sensitivity, quick reaction, and in situ features. FPs have been widely employed in biomedical studies spanning from imaging to medication delivery [142,143]. More significantly, they are biodegradable and may be eliminated from the body after their programmed activities have been completed. FPs are typically created by incorporating organic-conjugated fluorophores into the center of hydrophobic polymer. Traditional fluorescent probes suffer from aggregation-caused quenching effects, limiting their application at high concentrations or in nanoparticles. The concentration of loading fluorophore is an important factor in determining the overall brightness of the polymer formation. The fluorescence intensity of the polymer is directly dependent on the dye loading efficiency at low loading concentrations [144]. However, when the dye concentration increases, the dye molecules begin to agglomerate in the particle core, causing the dyes’ light emission to be quenched. AIE polymers have surmounted this challenge since the polymer molecules become extremely emissive when aggregated [145]. AIE bioconjugates, formed by covalently linking AIE luminogens to biomolecules, are particularly promising candidates for biomedical applications due to their excellent biocompatibility, good water solubility, high specificity to the target of interest, wide functionality, and smart responsiveness [146,147,148].

In this context, Liu et al. [149] (Figure 19) created a new photoactive polymer for light-controlled gene delivery that constituted an AIE photosensitizer and OEI coupled through a ROS-responsive linker. The polymer contained an AIE PS conjugated with oligoethyleneimine (OEI; 800 Da) via an aminoacrylate (AA), where PEG was further grafted to fine-tune the water solubility of the polymer. The polymer may self-assemble into nanoparticles (NPs) exhibiting intense red fluorescence for bioimaging in aqueous conditions.

When exposed to light (at a much lower light dose than in photodynamic therapy), the polymer vector may cause endo/lysosomal escape as well as DNA unpacking, allowing it to efficiently transfer DNA to the cytosol of cells while causing minor damage. When exposed to visible light, the produced ROS disrupted the endo/lysosomal membrane and the polymer, resulting in light-controlled endo/lysosomal escape and unpacking of DNA for effective gene delivery. Simultaneously, the ROS degrades the polymer and promotes the reversion of high molecular weight complexes to their low molecular weight counterparts, resulting in DNA unpacking. This research lays the groundwork for a potential light-controlled platform for simultaneous endo/lysosomal escape and DNA unpacking, both of which are required for effective gene delivery.

In the same context, Gao et al. [150] (Figure 20) fabricated mitochondria-targeted NPs with a high ROS quantum yield of 77.9% using a novel AIE crosslinked copolymer with FR/NIR subcellular bio-imaging capability. Cell viability experiments demonstrated that the polymer has high cytocompatibility in the dark but causes severe cytotoxicity in cancer cells when exposed to low-cost white light (10 mW cm^−2^). They fabricated a crosslinked copolymer containing N-(2-hydroxypropyl)methacrylamide (HPMA), 2-aminoethyl methacrylate (AEMA), and tri-phenyl-phosphonium (TPP) [151]; the latter is a well-established chemical scaffold that targets mitochondria. Within the mitochondria of live cells, cargo molecules covalently linked to TPP accumulate.

TPP and free TPP polymer NPs have 39.0 mV and 11.3 mV surface zeta potentials, respectively. TPP polymer NPs had a higher surface zeta potential, indicating that TPP was conjugated at the NP surface, satisfying the mitochondrial targeting criterion. The difference in cellular uptake between polymer with and without TPP was confirmed using flow cytometry, and the results demonstrated that the presence of TPP improved the cellular uptake.

AIE polymers have also been used as a drug delivery system (DDS) [152,153,154]. A DDS is a formulation or a device that allows the entrance of a therapeutic material into the body and increases its efficacy and safety by managing the rate, timing, and location of drug release in the body. The delivery of the therapeutic product, the release of the active chemicals by the product, and the subsequent transport of the active ingredients through biological membranes to the site of action are all part of this process [155]. Thus, local drug delivery is highly demanded in inoperable cancers and prevents local tumor recurrence. In terms of drug delivery, a self-indicting drug delivery system based on a novel aggregation-induced emission thermoresponsive hydrogel, based on PEG, poly(propylene glycol) (PPG), and tetraphenylethene (TPE), featuring AIE that was be tuned depending on Doxorubicin (Dox) concentration and temperature was described by Loh et al. [156] (Figure 21). The hydrogel accurately monitored in vivo drug release using a self-detecting device. It is simple to track medication depletion and reinject to maintain a dose within the ideal therapeutic window.

Due to difficulties in penetrating deep tissue and skin, the AIE thermogel was injectable in solutions at room temperature, forming depots containing Dox at the injection site, as observed by the signals generated by gel aggregation, and were collected at 515–575 nm. Because TPE aggregation was restored as Dox was released over time, the EPT thermal photoluminescence signal increased in a dose-dependent manner. The self-indicating EPT thermogel is useful because the signals from the matrix showed the precise release of Dox depending on the change in Dox concentration. These findings suggest that when chemotherapeutics is used, the delayed release of drug-encapsulated micelles from AIE thermogel may help to limit tumor growth. In addition, delayed release causes less harm in mice.

In regards of DDS, Tang and colleagues [157] created a conjugated polymer (PTB-APFB) containing benzothiadiazole and tetraphenylethene (Figure 22) that has excellent ROS-generation ability and selectivity for pathogenic microbes over mammalian cells.

Due to its D-π-A structure and AIE property, the polymer has a significant ROS production ability in the aggregate state under the light. PTB-APFB has excellent selectivity for microbes over mammalian cells. Because of its balanced hydrophilicity and hydrophobicity, PTB-APFB exhibited excellent selectivity for microbes over mammalian cells. In vitro and in vivo antibacterial studies demonstrate that PTB-APFB can effectively inhibit bacterial growth and accelerate the healing process. Additionally, in vitro and in vivo research suggests that PTB-APFB is biocompatible. As a result, both preclinical and clinical trials indicate that PTB-APFB holds considerable promise as an antibacterial agent (Figure 23).

Bacterial growth inhibition was confirmed in vitro and in vivo after polymer treatment under light irradiation. Notably, after therapy with their polymer, infection recovery is faster than after treatment with cephalothin [158]. As a result, in real-world applications, this polymer holds a lot of promise for treating bacteria-related diseases.

## 4. Conclusions and Future Outlook

This review focused on research conducted over the last six years, specifically on the design and preparation of fluorophores containing polymers and AIE FPs. In the first, researchers have taken an interest in fluorophores-containing polymers due to their distinct photoluminescence, tunable emission colors (including white light emission), and biocompatibility, despite the fact that they are not commercially viable at the moment. The development of stimuli response fluorescent material is an exciting field, particularly in the study of cells, where changes in pH or temperature can be monitored during drug administration to better understand how small molecules act in normal or cancerous cells. Aside from improving the design of signal transduction systems in sensors and biosensors, the electrochromic properties of conducting polymers can also be used to improve their performance.

Conjugated polymers have undoubtedly been utilized to fabricate flexible optoelectronic devices and photo transistors because to their softness, resilience, and light weight, but have received less attention in the fabrication of a portable electrochemical sensor and in electrochemical applications. Thus, the tunable features of polymers, such as band gap tuning, might be advantageous in electrochemical applications. Additionally, although some conjugated frameworks have been utilized to deliver pharmaceuticals such as doxorubicin and for bio imaging, their biological uses are still in their infancy. While designing these materials, several critical factors should be considered, including bioavailability, biocompatibility, solubility in aqueous and biological environments, and low cytotoxicity. Without a doubt, developing new materials will require the collaborative efforts of chemists, biologists, and physicists in order to achieve higher quantum yield and stability for advanced applications. These materials can emit light with great efficiency in dilute solutions; however, their fluorescence is diminished or perhaps eliminated in concentrated solution or solid form, contrary to AIE polymers.

AIE polymers have achieved extraordinary success over the last two decades, resulting in additional research and discoveries with enormous potential in this field. It is desirable to develop and prepare novel AIE-active monomers in order to create innovative AIE polymers with novel properties and functionalities. Persistent efforts are required to develop novel and efficient synthetic techniques for AIE polymerization, with a particular emphasis on metal-free and spontaneous polymerization processes. The in situ synthesis of AIE polymers from inactive AIE monomers will be an intriguing direction to pursue. Additionally, AIE polymers with controlled molecular weights, repeated unit sequences, and well-defined architectures are in high demand. To meet the demands of various research frontiers, additional AIE-active polymers with variable architectures and multifunctional capabilities should be developed. For instance, visualizing the synthesis or aggregation process may aid in testing and validating established polymer science principles. Although the use of AIE polymers in polymer light-emitting diodes has been investigated, the devices’ efficiency still requires improvement. Furthermore, photochemically responsive systems are still in their infancy. Despite the fact that hundreds of photochemically responsive units have been identified, only a few have been successfully used in the fabrication of photochemically responsive AIE polymers. By discovering novel photo-responsive units that operate via distinct mechanisms, such as catalyst-free light-controlled cycloaddition, we can significantly diversify the types and accelerate the development of photo-chemical responsive systems.

We anticipate that AIEgens will play critical roles in the discovery of new photo responsive polymers by integrating them with diverse production strategies and cutting-edge methodologies. We believe that this review will encourage additional scientific researchers to work in this promising field, as well as foster a knowledge of the full potential and further development of classic polymers and AIE polymers for wider applications. As the adage goes, “Rome was not built in a day”.

## Figures and Tables

**Figure 1 polymers-14-01118-f001:**
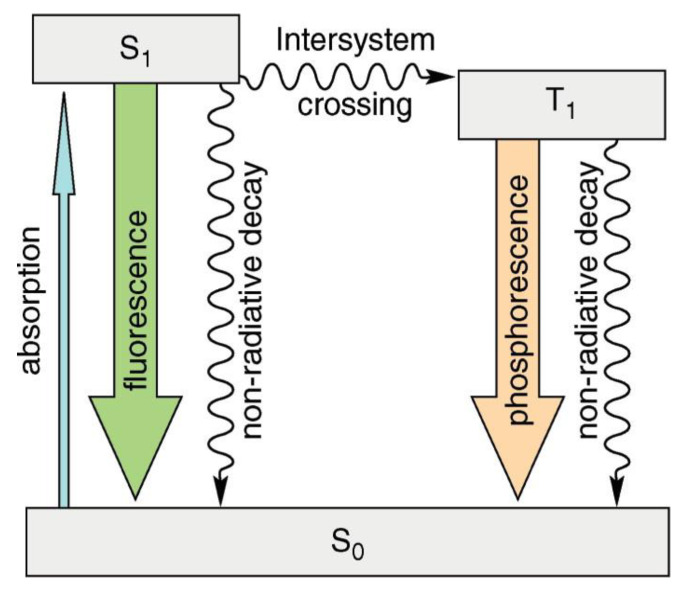
Simplified diagram (Perrin–Jablonski) showing the difference between fluorescence and phosphorescence. Reproduced with permission from reference [7]. Copyright 2011, American Chemical Society.

**Figure 2 polymers-14-01118-f002:**
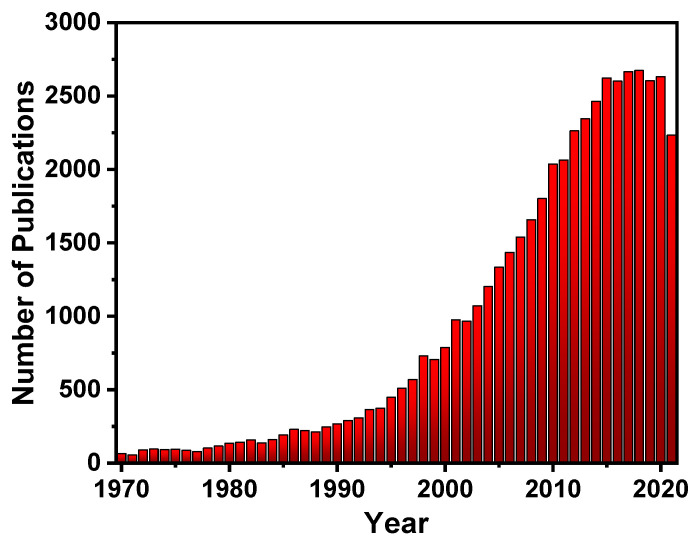
The number of publications per year on (FPs) (obtained from Scifinder^n^).

**Figure 3 polymers-14-01118-f003:**
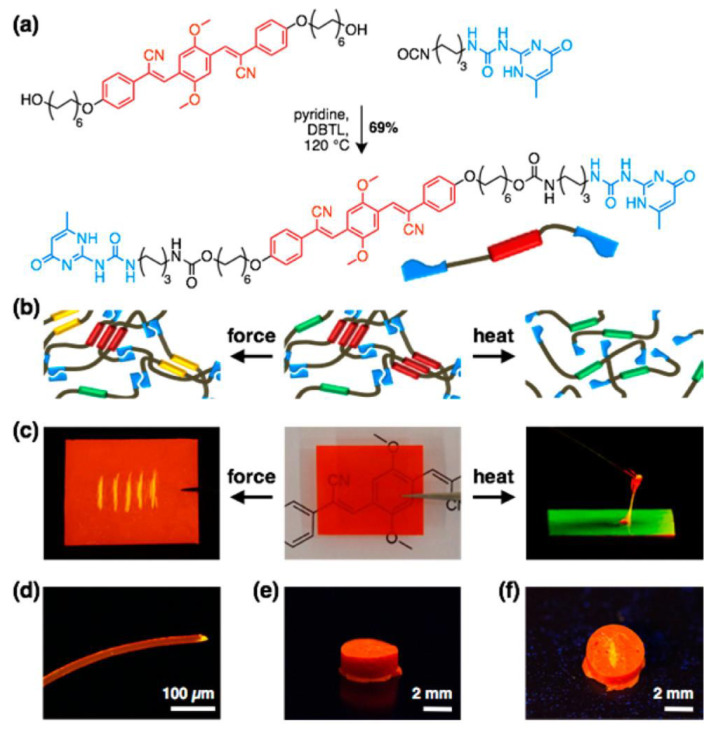
(**a**) Synthesis of the UPy-functionalized cyano-OPV. (**b**) Schematic of supramolecular assemblies. (**c**) Pictures of films made from the UPy-functionalized cyano-OPV, illustrating mechano- (left) and thermoresponsive (right) luminescent behavior. The fluorescence changes from red to yellow upon scratching (left). Upon heating (180 °C), a viscous green-light-emitting fluid is formed, which solidifies into a red-light-emitting solid when cooled (right). (**d**) Fluorescence microscopy image of a fiber made from 2; note the yellow fluorescing severed edge. (**e,f**) Photographs of a cylinder made from the UPy-functionalized cyano-OPV before (**e**) and after (**f**) scratching its surface. Images displaying fluorescence were recorded under illumination with 365 nm UV light. Adapted with permission from reference [69]. Copyright 2017, American Chemical Society.

**Figure 4 polymers-14-01118-f004:**
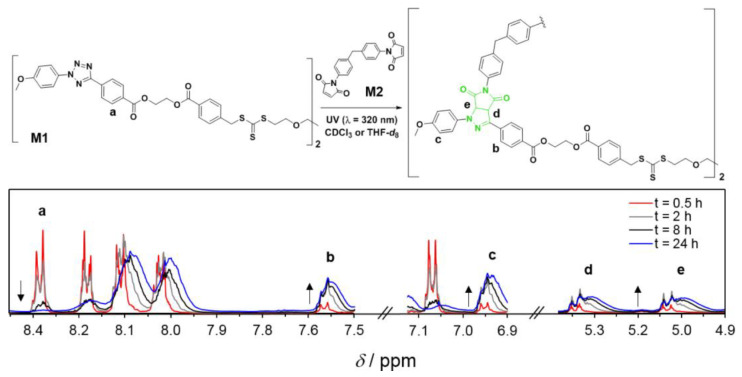
General reaction scheme for the UV-initiated (λ_max_ = 320 nm) step-growth polymerization of monomers M1 (tetrazole RAFT agent) and M2 (bismaleimide), [M1]_0_ = [M2]_0_ = 50 mmol L^–1^, in CDCl_3_ or THF-d_8_. The ^1^H NMR spectra (CDCl_3_) display the evolution of the signals used to determine the monomer conversion and the pyrazoline yield. Reproduced from reference [73].

**Figure 5 polymers-14-01118-f005:**
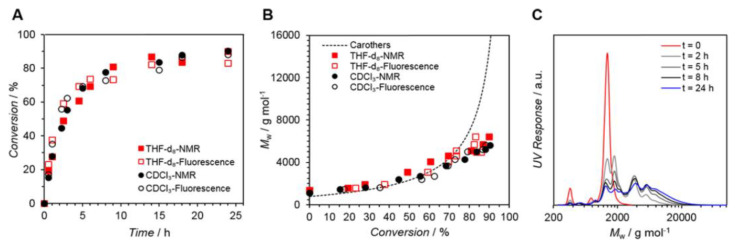
(**A**) Kinetic plot displaying conversion vs. reaction time for the polymerization of M1,2 (Solvents = THF-d8 or CDCl_3_. NMR (solid symbols) and fluorescence (open symbols) determined the conversion. (**B**) Corresponding Mw vs. conversion plot (Carothers curve represented as a dotted line). (**C**) Mw values determined via SEC analysis. Reproduced with permission from reference [73]. Copyright 2017, American Chemical Society.

**Figure 6 polymers-14-01118-f006:**
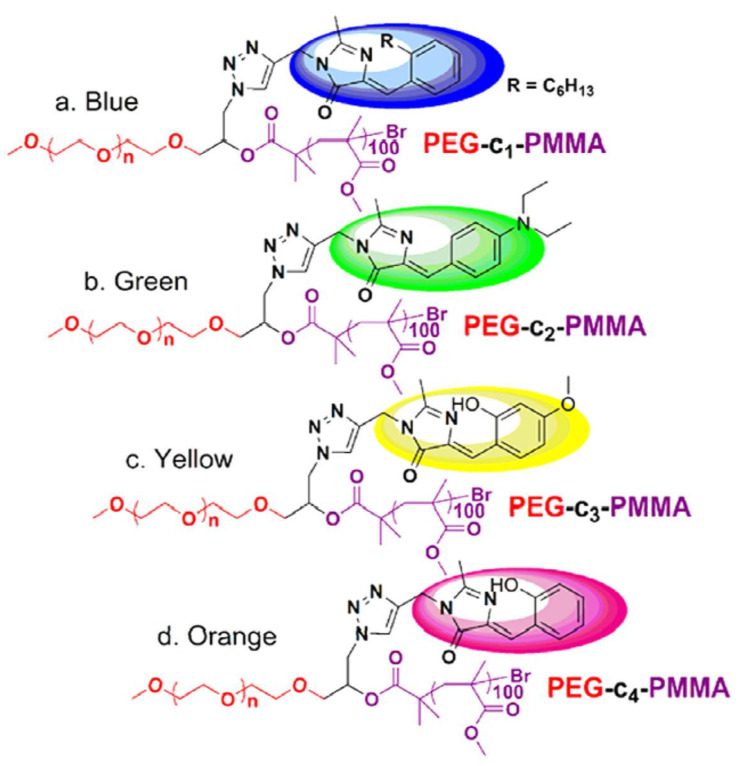
Polymer structure and their fluorescent color dependence on the fluorophore attached: (**a**) blue, (**b**) green, (**c**) yellow, and (**d**) orange. Adapted with permission from reference [74]. Copyright 2015, American Chemical Society.

**Figure 7 polymers-14-01118-f007:**
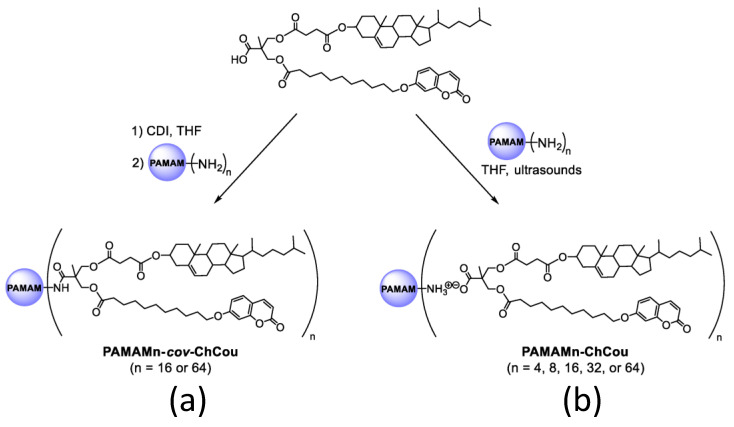
Covalent (**a**) and ionic (**b**) functionalization of PAMAM dendrimer with a hybrid bisMPA dendron bearing cholesterol and coumarin moieties. Reproduced from reference [79]. Licensed under CC-BY 4.0.

**Figure 8 polymers-14-01118-f008:**
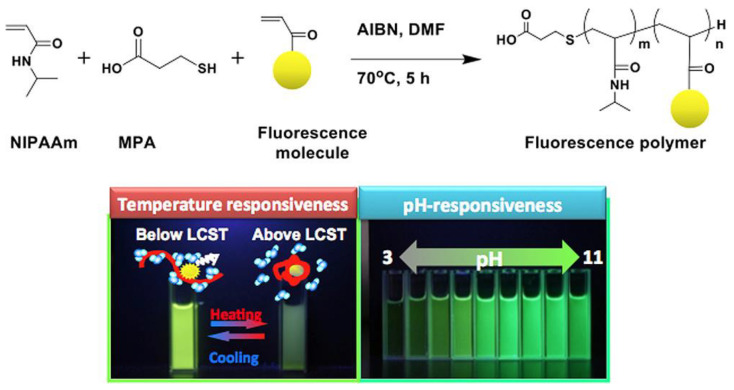
General synthetic scheme for the temperature and pH-responsiveness poly-NIPAAm. Reproduced from reference [82].

**Figure 9 polymers-14-01118-f009:**
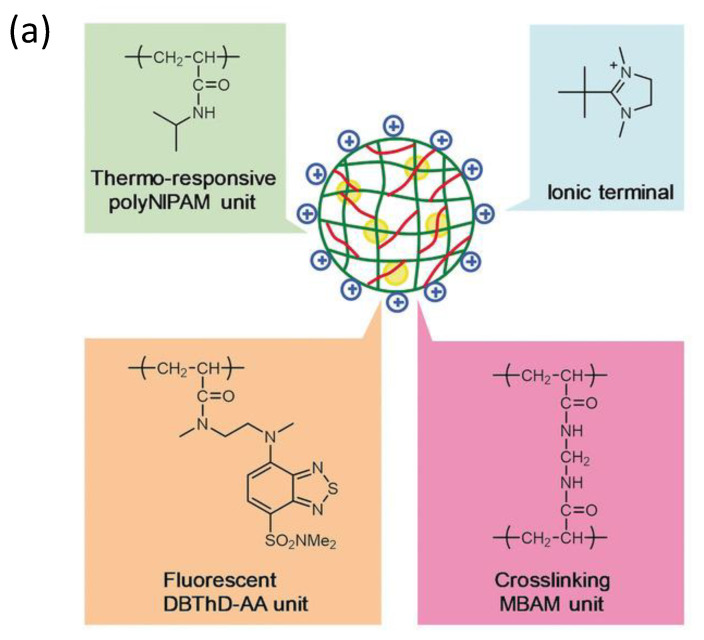
(**a**) Chemical structure of the CFNT. At lower temperatures, the CFNT swells by absorbing water into its interior, where the environment-sensitive DBThD-AA units are quenched by neighboring water molecules. When heated, CFNT shrinks with the release of water molecules, resulting in fluorescence from the DBThD-AA units. (**b**) Differential interference contrast (DIC) image, confocal fluorescence image, and merged image of HeLa cells treated with 1 (0.05 *w*/*v*%), and (**c**) DIC image, confocal fluorescence image, and merged image of MOLT-4 cells treated with the CFNT (0.05 *w*/*v*%). Reproduced with permission from reference [87]. Copyright 2018, John Wiley and Sons.

**Figure 10 polymers-14-01118-f010:**
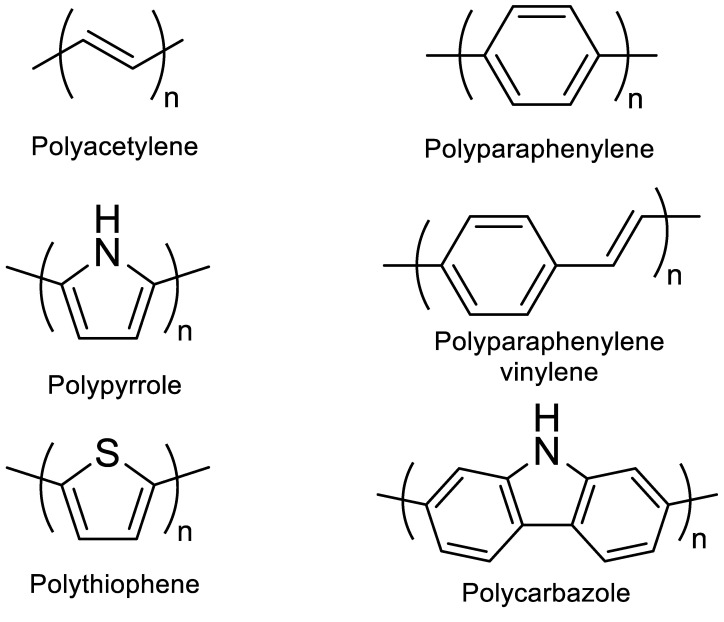
Classic examples of conducting polymers.

**Figure 11 polymers-14-01118-f011:**
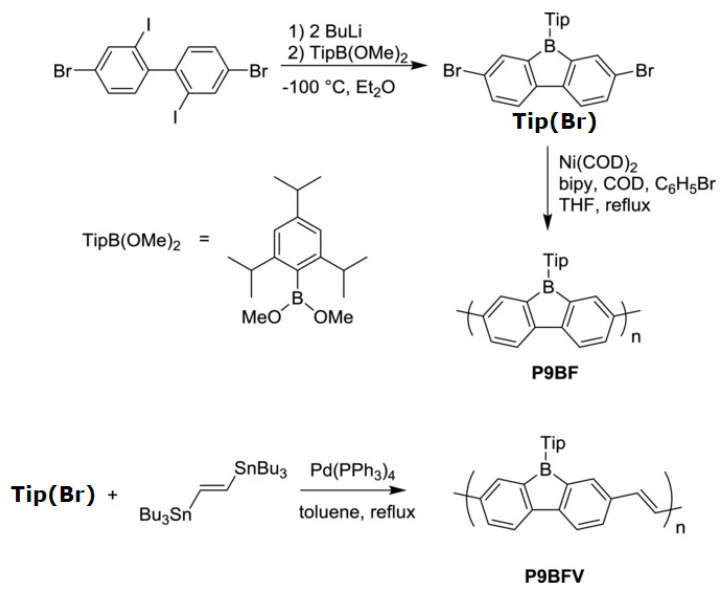
Synthesis of the polymer P9BFV. Reproduced with permission from reference [95]. Copyright 2015, John Wiley and Sons.

**Figure 12 polymers-14-01118-f012:**
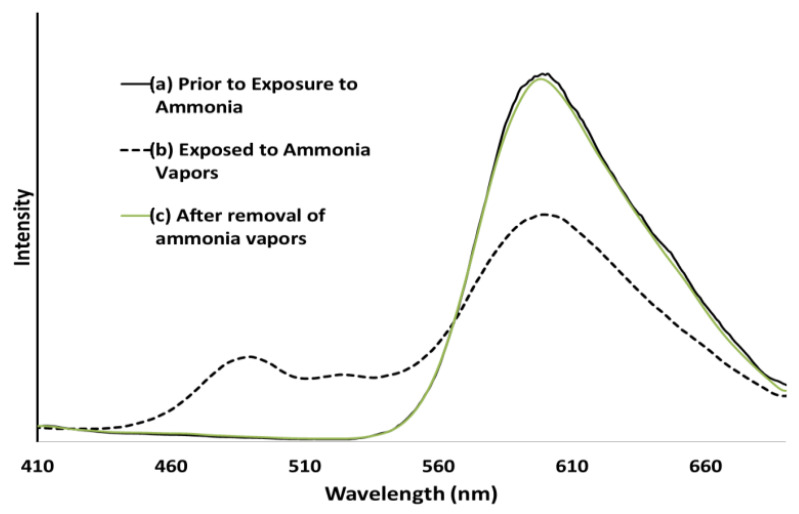
(**a**) Emission spectra of a P9BFV film before exposure to NH_3_ vapors, (**b**) emission spectra of a P9BFV film while being exposed to NH_3_ from aqueous ammonium hydroxide, (**c**) emission spectra of a P9BFV film 5 min after removal of the NH_3_ source and NH_3_ vapors. Reproduced with permission from reference [95]. Copyright 2015, John Wiley and Sons.

**Figure 13 polymers-14-01118-f013:**
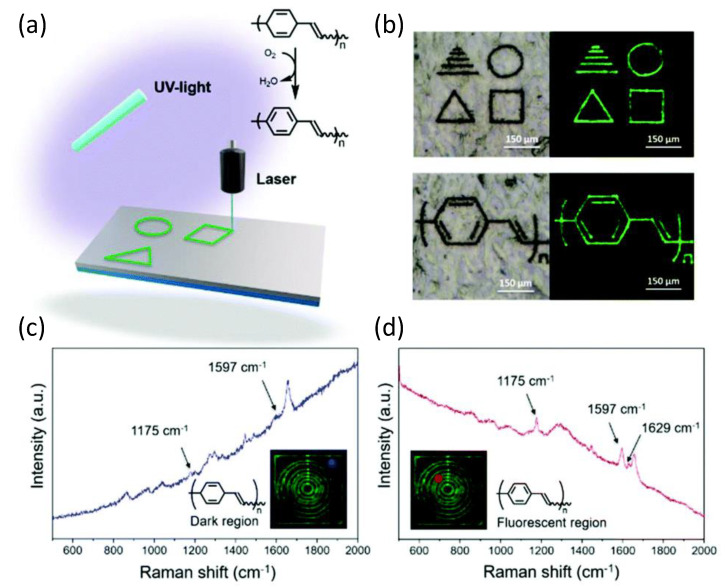
(**a**) Laser-induced oxidation of poly(barrelene) to polyphenylene vinylene (PPV). (**b**) Examples of patterns that were written. Photographs were taken under visible light (left) and UV light (λex = 488 nm; right). (**c**,**d**) Raman spectra were recorded for the areas indicated in blue or red (see insets) of a film subjected to DLW. The films were prepared by drop-casting a solution of poly(barrelene-co-norbornene) (200 mg) in CH_2_Cl_2_ (4 mL) into a PTFE mold. Reproduced from reference [101] with permission from the Royal Society of Chemistry (Licensed under CC-BY 3.0).

**Figure 14 polymers-14-01118-f014:**
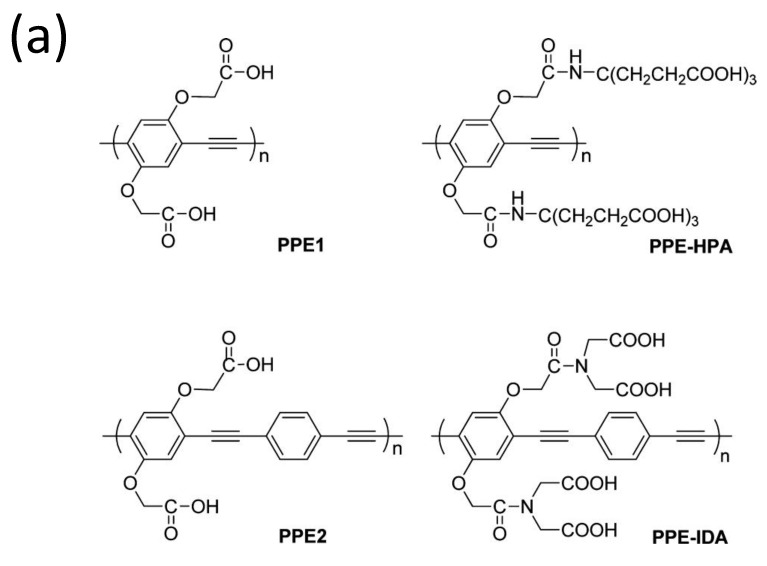
(**a**) Structures of the four conjugated polyelectrolytes, PPE1, PPE2, PPE−IDA, and PPE−HPA, (**b**) response patterns constructed based on fluorescence quenching of the four polymers by eight metal ions at 5 μM each. The response patterns were generated from the ratios of the polymers’ initial to final emission intensities. Error bars represent the standard deviations of six replicates for each PPE−metal ion pair. Polymers are PPE1, PPE2, PPE−IDA, and PPE−HPA, and metal ions are Pb^2+^, Hg^2+^, Fe^3+^, Cr^3+^, Cu^2+^, Mn^2+^, Ni^2+^, and Co^2+^. Adapted with permission from reference [117]. Copyright 2015, American Chemical Society.

**Figure 15 polymers-14-01118-f015:**
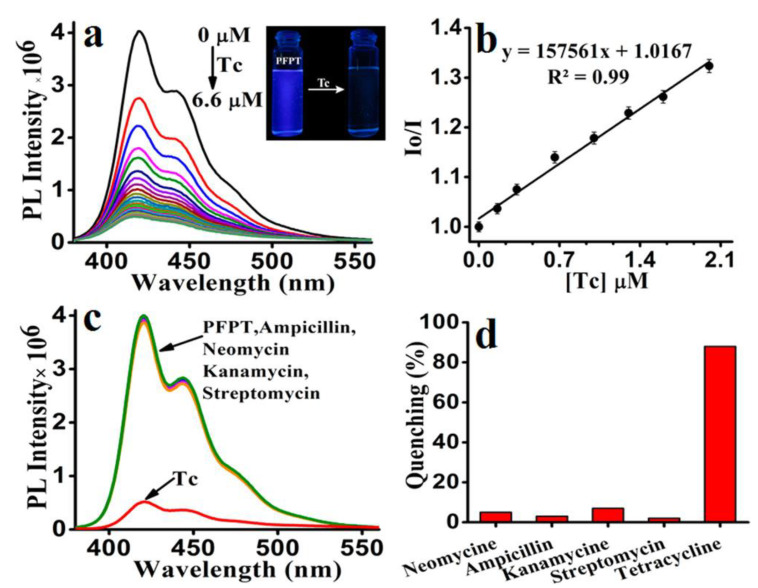
(**a**) Fluorescence spectra of PFPT (6.6 μM) with increasing concentration of Tc (6.6 μM) in aqueous media (HEPES 10 mM, pH 7.4). Inset: color change of PFPT under UV light (lamp excited at 365 nm) before and after adding Tc. (**b**) Stern–Volmer plot of PFPT upon the addition of Tc in aqueous media. Effect of Tc (6.6 μM) and various other antibiotics (6.6 μM) on the (**c**) emission spectra of PFPT and their corresponding (**d**) bar diagram depicting the quenching percentage. Reproduced with permission from reference [121]. Copyright 2017, American Chemical Society.

**Figure 16 polymers-14-01118-f016:**
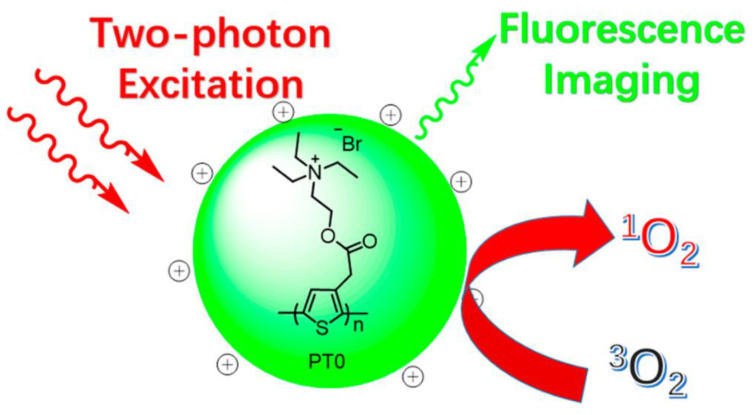
Illustration of PT0 for simultaneous two-photon excitation fluorescence imaging and photodynamic therapy. Reproduced with permission from reference [127]. Copyright 2017, American Chemical Society.

**Figure 17 polymers-14-01118-f017:**
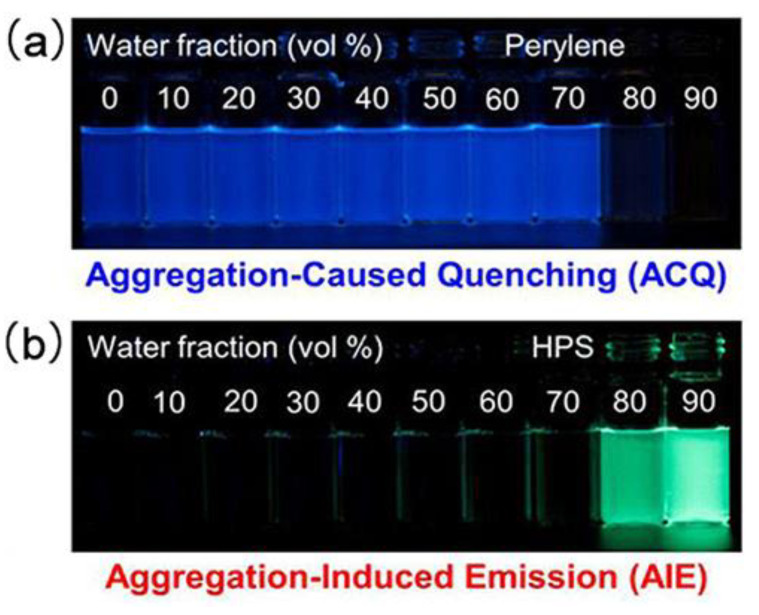
Fluorescence photographs of solutions/suspensions of (**a**) ACQ effect of perylene (20 μmol/L) and (**b**) AIE effect hexaphenylsilole (20 μmol/L) in THF-water mixtures with increasing water contents fraction. Adapted with permission from reference [134]. Copyright 2015, American Chemical Society.

**Figure 18 polymers-14-01118-f018:**
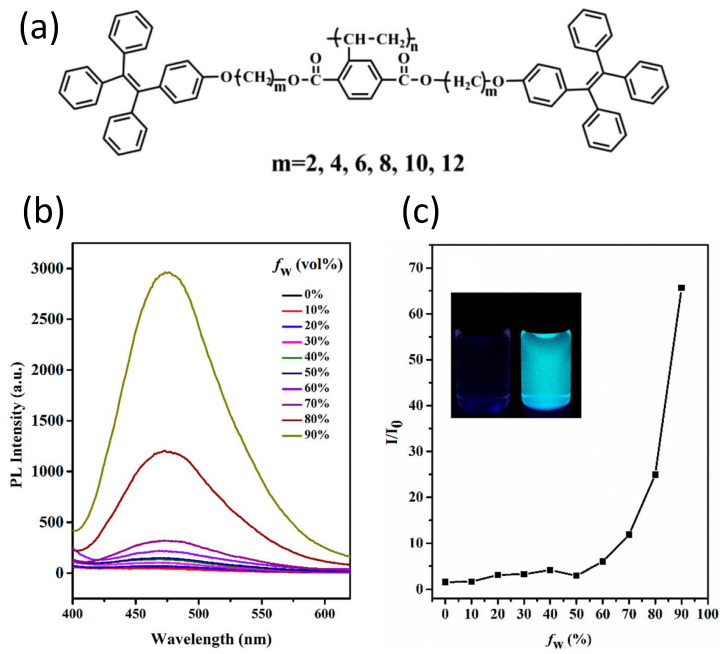
(**a**) Molecular structures of the LCPs. (**b**) Emission spectra of the polymer (shown above, m = 6) in THF/H_2_O mixtures with various water fractions. (**c**) Plots of I/I_0_ values versus water contents of the aqueous mixtures at 476 nm for the polymer (show above, m = 6); the inset photos of THF with 0 and 90% water fractions taken under UV light. Reproduced with permission from reference [140]. Copyright 2017, American Chemical Society.

**Figure 19 polymers-14-01118-f019:**
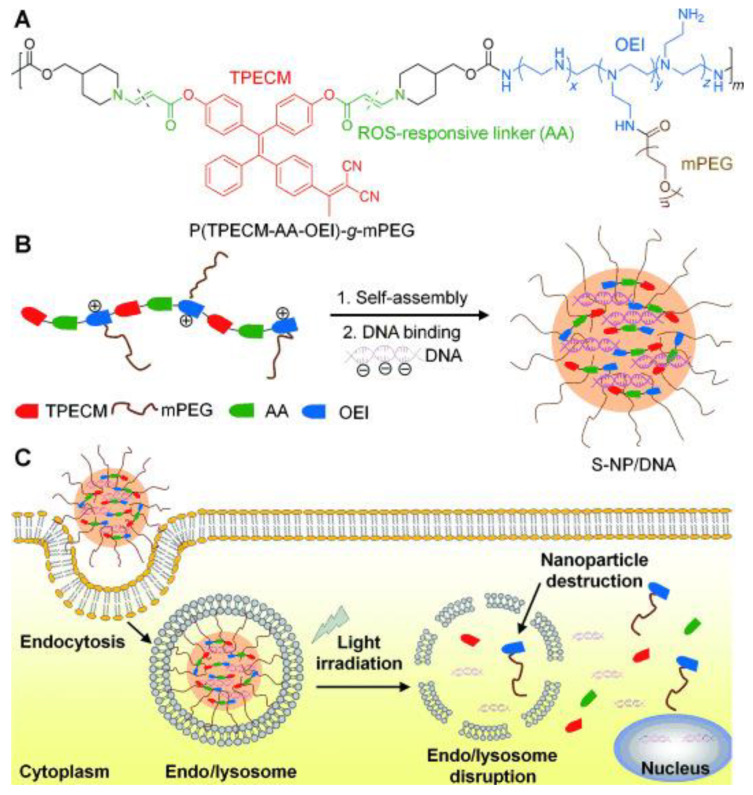
(**A**) Chemical structure of the ROS-responsive polymer P(TPECM-AA-OEI)-g-mPEG. (**B**) ROS-sensitive nanoparticles (S-NPs) self-assembled from P(TPECM-AA-OEI)-g-mPEG in aqueous media and their complexation with DNA to form S-NPs/DNA. (**C**) The itinerary of S-NPs/DNA to the transgene expression. S-NPs/DNA was endocytosed by the cells and entrapped in endo/lysosomes. Upon light irradiation, the generated ROS can concurrently destroy the endo/lysosomal membrane to facilitate the vector escape and break the S-NPs to favor DNA unpacking, leading to DNA release for nuclear entry and transcription. Reproduced with permission from reference [149]. Copyright 2015, John Wiley and Sons.

**Figure 20 polymers-14-01118-f020:**
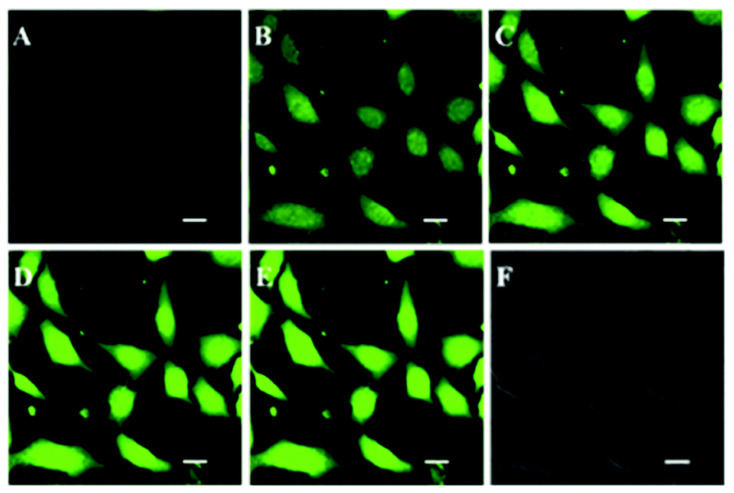
Confocal laser scanning microscopy images of A549 cancer cells after incubation with the AIE polymer containing TPP (40 μg mL^−1^) and dichlorofluorescein diacetate (10 μM) under white light irradiation for (**A**) 0 min, (**B**) 2 min, (**C**) 4 min, (**D**) 6 min, and (**E**) 8 min. (**F**) is the bright-field image. λ_ex_ = 488 nm, λ_em_ = 505–525 nm. The scale bar is 20 μm. Reproduced from reference [150] with permission from the Royal Society of Chemistry.

**Figure 21 polymers-14-01118-f021:**
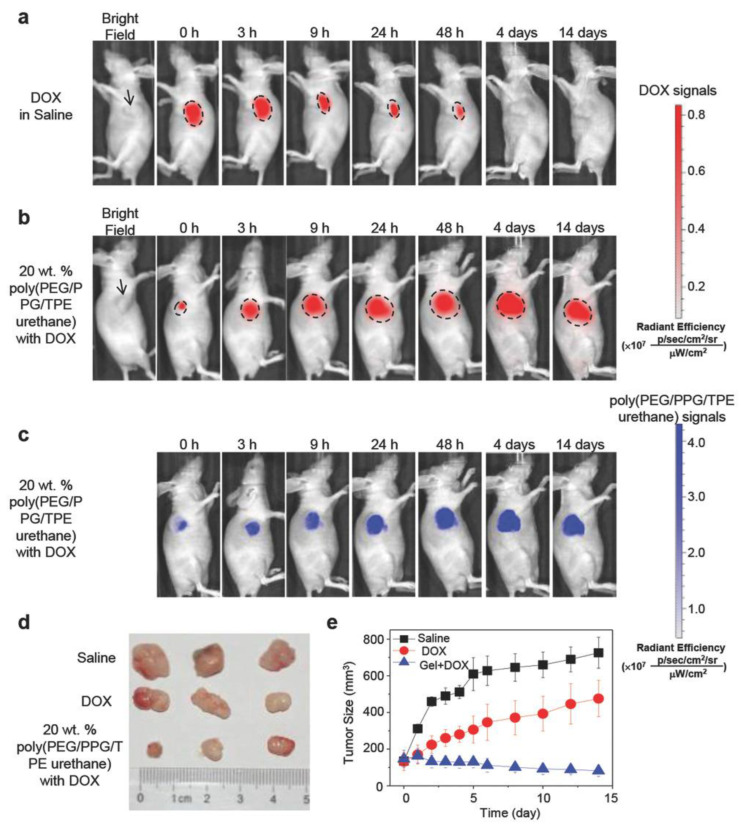
In vivo mouse model. (**a**–**c**) In vivo noninvasive fluorescence imaging of nude mice bearing HepG2 tumor with postintratumoral injection of 50 µL DOX in saline or DOX loaded thermogel for indicated time points. The black arrow indicated the location of the tumor. (**d,e**) Inhibition of tumor volume by intratumoral injection of saline, DOX, or DOX loaded EPT thermogels. Dorsal subcutaneous implantation of HepG2 cancer cells into mice was followed by administration of each solution after tumors had reached a volume of ~150 mm^3^. The excised tumor removed after 14 d was taken for evaluation. Reproduced with permission from Reference [156]. Copyright 2016, John Wiley and Sons.

**Figure 22 polymers-14-01118-f022:**
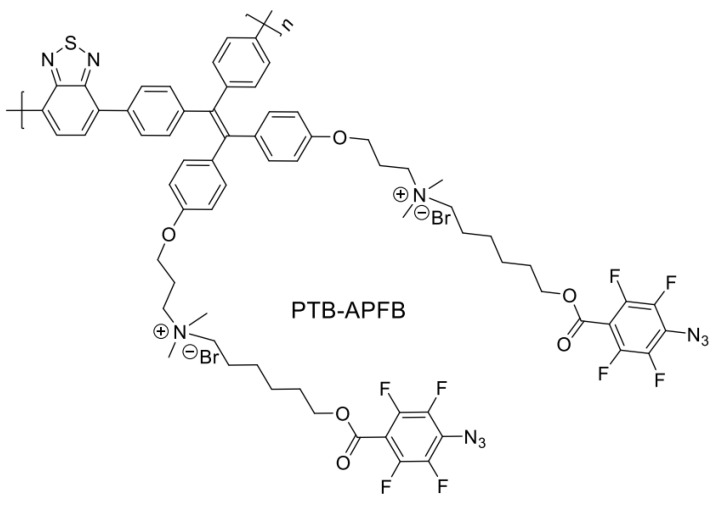
Structure of the PTB-APFB polymer. Adapted with permission from reference [157]. Copyright 2020, John Wiley and Sons.

**Figure 23 polymers-14-01118-f023:**
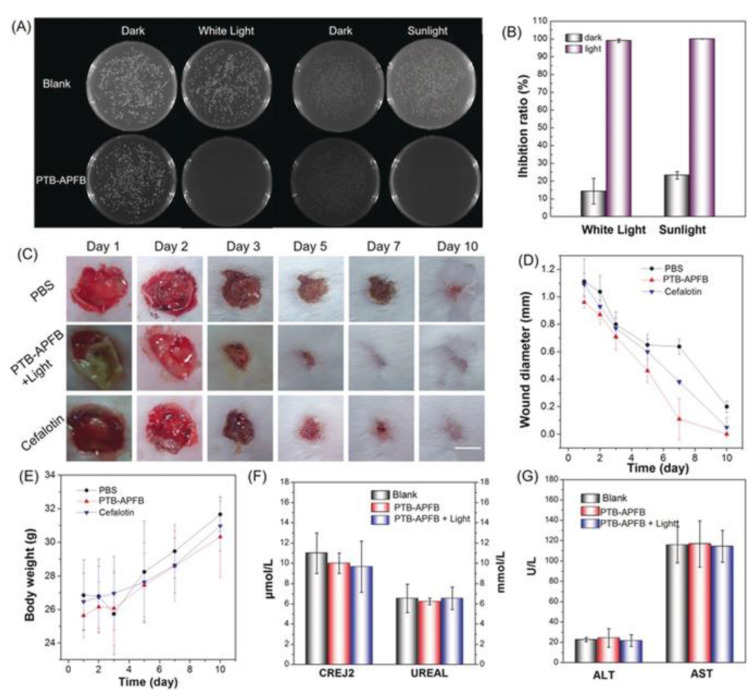
(**A**) Photographs of NB agar plate and (**B**) biocidal activity of *S. aureus* with PTB-APFB treatment in dark and under white light and sunlight. (**C**) Photographs of the *S. aureus*-infected skin of mice during treatment with the different formulations, and (**D**) the size of the infected area as well as (**E**) the body weights of the mice. Scale bar = 1 cm. (**F**) Levels of CREJ2 and UREAL (biomarkers), and (**G**) ALT and AST (liver function biomarkers) in blood samples from mice with different treatments. Reproduced with permission from reference [157]. Copyright 2020, John Wiley and Sons.

**Table 1 polymers-14-01118-t001:** Characterization of synthetic FPs. Reproduced from reference [82].

Polymer	M_w_ ^a^	Đ	λ_ex_ (nm)	λ_em_ (nm)
P(NIPAAm-co-FL)	52,994	1.979	490	515
P(NIPAAm-co-CO)	47,494	1.873	376	460
P(NIPAAm-co-RH)	45,905	2.028	540	588
P(NIPAAm-co-DA)	43,596	1.984	335	526

^a^ Determined by GPC using DMF with 10 mM LiCl.

## Data Availability

No new data were created or analyzed in this study. Data sharing is not applicable to this article.

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
