# Peer review of "Fluorescent Polymers Conspectus"

_polymers, 2022, doi:10.3390/polym14061118_

Round 1

Reviewer 1 Report

In this submitted manuscript, Dr. Ahumada and Magdalena Borkowska reviewed the recent developments of fluorescent polymers (FPs), which include both the two branches of macromolecules containing fluorophores in their structures and aggregation-induced emission fluorescent polymers.

The authors systematically summarized the history of fluorescent polymers, and then discussed them in detail in the aspects of the non-conjugated and conjugated polymers containing fluorophores, and aggregation-induced emission macromolecules. Their applications in sensing and bioimaging were also discussed.

The manuscript is written in high quality with plentiful discussions in fluorescent polymers design, preparation, characterization, and applications. In terms of the manuscript content and importance, the work could appeal to the broad readership of Polymers and I would recommend accepting it for publication after minor revisions.

The authors have already nicely explained and discussed the comments I gave in the first review and made good revisions thereafter.

The minor things that need to be revised further are when the short name appears for the first time, its full name should be given. On lines 13 and 21, the full names of FPs and AIEgens should be provided, and they should be revised as "fluorescent polymers (FPs)" and "aggregation-induced-emission-based fluorescent materials (AIEgens)".

Author Response

Comment: In this submitted manuscript, Dr. Ahumada and Magdalena Borkowska reviewed the recent developments of fluorescent polymers (FPs), which include both the two branches of macromolecules containing fluorophores in their structures and aggregation-induced emission fluorescent polymers. The authors systematically summarized the history of fluorescent polymers, and then discussed them in detail in the aspects of the non-conjugated and conjugated polymers containing fluorophores, and aggregation-induced emission macromolecules. Their applications in sensing and bioimaging were also discussed. The manuscript is written in high quality with plentiful discussions in fluorescent polymers design, preparation, characterization, and applications. In terms of the manuscript content and importance, the work could appeal to the broad readership of Polymers and I would recommend accepting it for publication after minor revisions. The authors have already nicely explained and discussed the comments I gave in the first review and made good revisions thereafter.

Reply: We appreciate reviewer taking the time to evaluate our paper and provide useful feedback. We have read your comments carefully and tried our best to address them one by one. Reviewer important and informative feedback has resulted in possible changes in the current edition. He/she raised a minor note which we now addressed.

Comment: The minor things that need to be revised further are when the short name appears for the first time, its full name should be given. On lines 13 and 21, the full names of FPs and AIEgens should be provided, and they should be revised as "fluorescent polymers (FPs)" and "aggregation-induced-emission-based fluorescent materials (AIEgens)".

Reply: We appreciate the reviewer's comments. The abstract has been revised, accordingly.

Reviewer 2 Report

The revised manuscript is significantly improved. The authors provide a critical analysis of the topic and insights about the remaining challenges and future directions for the field. The authors refer to the latest reports on the fluorescent polymer. Thus, I think the presented review is suitable for publication in Polymer journal.

Author Response

Comment: The revised manuscript is significantly improved. The authors provide a critical analysis of the topic and insights about the remaining challenges and future directions for the field. The authors refer to the latest reports on the fluorescent polymer. Thus, I think the presented review is suitable for publication in Polymer journal.

Reply: We thank reviewer comments. We are pleased the Reviewer found this work novel and of interest.

Reviewer 3 Report

I've noted that the manuscript has undergone a massive round of revision by the authors taking into account the comments and inputs from both reviewers. The effort in improving the quality and content of the manuscript is commendable and the reviewer noted significant improvement to the work. I therefore opine that the manuscript may be accepted as it is. If the authors still wish to further improve on the work, I feel perhaps the discussions revolving around Figure 2 (on increasing yearly publications in FP) may not be necessary, and that the introduction on conjugated polymers (Page 9) can be further summarized into 1 paragraph.

Author Response

Comment: I've noted that the manuscript has undergone a massive round of revision by the authors taking into account the comments and inputs from both reviewers. The effort in improving the quality and content of the manuscript is commendable and the reviewer noted significant improvement to the work. I therefore opine that the manuscript may be accepted as it is.

Reply: We appreciate the reviewer's effort in evaluating our article and providing constructive criticism. We have attentively reviewed your remarks and have attempted to address them one by one. Significant and informative feedback from reviewers has resulted in possible revisions to the current edition. He/she brought up a small point, which we respond here.

Comment: If the authors still wish to further improve on the work, I feel perhaps the discussions revolving around Figure 2 (on increasing yearly publications in FP) may not be necessary, and that the introduction on conjugated polymers (Page 9) can be further summarized into 1 paragraph.

Reply: We appreciate the feedback, but we'd like to keep the text as is (without major modifications).

This manuscript is a resubmission of an earlier submission. The following is a list of the peer review reports and author responses from that submission.

Round 1

Reviewer 1 Report

Ahumada and Borkowska presented this review article on recent advances in fluorescent polymers, in which they discuss selected examples of fluorescent polymers, from structure to synthesis and applications, based on three different categories: (1) Non-conjugated polymers containing fluorophores; (2) conjugated polymers containing fluorophores; and (3) AIE polymers, after a brief introduction on the nature and mechanism of fluorescence, a brief overview of conjugated polymers, and AIE materials. The review ends with a short conclusion and perspective on what the authors believed are key challenges in this area, and how this review may benefit research community at large. 

Overall, I opine that the article fails to provide a holistic and in-depth discussion on the broad scope of content covered by the proposed title "Recent advances in fluorescent polymers". Perhaps the title is too general and broad, and thus the structure and content of this manuscript seem a little over-stretched, trying to touch a bit of every possible parts of the topic, but ends up not being able to provide a robust discussion for each part, in total. The structure of the review - discussing a few examples of each type of non-conjugated, conjugated and AIE fluorescent polymers - makes the article appear rather patchy, without coherent flow or discussion between sections. I would suggest the authors rather focus on just one of the three areas and provide a more robust and detailed discussion and analysis for that sub-topic on fluorescent polymers. The introduction and conclusion also needs improvement as the flow of content is rather fragmentary, e.g. with the sudden switch of discussion from applications conjugated polymers to AIE, without any link in between.

I therefore opined that the review needs to be re-written and re-structured in order to be acceptable for publication. The review does not add much value to what we have already knew and it fails to address gaps that exists that are not addressed by other similar reviews. In addition to the general points raised above, the following are some other points that require addressing:

(1) The language needs to be improved. Many of the sentences have syntax error or are structured awkwardly. There are many sentences which meaning are not clear as well. For e.g. the abstract "In the latter, AIE FPs take advantage of a novel mode of fluorescence that depends on the material's order". 

(2) The introduction definitely needs to be revised. There is a lack of flow of thoughts between paragraphs from first mentioning the mechanism of fluorescence, to then types and applications of fluorescent compounds to conjugated polymers' applications in optoelectronics and then to AIE. Contents in individual paragraph appear to just be conveniently mentioned or regurgitation of known-facts without any actual value adding to building the stage for the rest of content discussed in the manuscript. 

(3) I fail to see the significance or importance of many of the examples discussed, with respect to other reports of FL polymers, i.e. why are these examples selected for discussion over others? Are they randomly chosen? They appeared to be discussed individually as isolated examples without contributing much to the understanding of the overall topic or sub-topic, making the content in each section appears rather fragmented. Furthermore, the author should consider providing an introductory paragraph for each section to broadly introduce the type of FL polymers e.g. general structures, general synthesis and design approach, general applications, etc., before discussing the different examples. 

(4) The authors should consider discussing how these recent reports on FL polymers mentioned serve as an improvement or breakthrough to older reports of FL polymers in terms of FL properties, or structure design novelty or even application performances, and not just summarizing each isolated report as another example of this type of polymer. 

(5) The "conclusion and future outlook" failed to provide a holistic perspective on future opportunities and current challenges facing the topic of FL polymers. For example, the authors mentioned "the future challenges will include using metal-containing polymers and the formulation of composites" but fail to substantiate why these areas are identified. 

Overall, I strongly suggest the authors to consider improving the structure and coherence of the review, the language and sentence structures, and with more in-depth discussion of the introduction, examples and outlook.

Reviewer 2 Report

“Review” should provide concise and precise updates on the latest progress made in a given area of research. It should also provide a critical analysis of the topic and insights about the remaining challenges and future directions for the field. The authors are too general in discussing the topic. In particular, there is no need to provide the principles of luminescence in the Introduction part. The authors do not refer to the latest reports on the fluorescent polymer. There are insufficient references to articles from the past five years

Thus, I do not think the presented review is not suitable for publication in Polymer journal.

Reviewer 3 Report

In this submitted manuscript, Dr. Ahumada and Magdalena Borkowska reviewed the recent developments of fluorescent polymers (FPs), which include both the two branches of macromolecules containing fluorophores in their structures and aggregation-induced emission fluorescent polymers.

The authors systematically summarized the history of fluorescent polymers, and then discussed them in detail in the aspects of the non-conjugated and conjugated polymers containing fluorophores, and aggregation-induced emission macromolecules.

The manuscript is written in high quality with plentiful discussions in fluorescent polymers design, preparation, characterization, and applications. In terms of the manuscript content and importance, the work could appeal to the broad readership of Polymers and I would recommend accepting it for publication after minor revisions.

  1. The abbreviation of fluorescent polymers, FPs, first appears on line 10. However, its full name was shown on line 17, and it would make more sense to give its full name (fluorescent polymers) on line 10 when its abbreviation appears for the first time.
  2. It was noticed that in the manuscript, the word “figure” was not capitalized in the first letter, as shown on lines 32, 53, 70, 101, 134, 141, 156, 184, 201, 226, 249, 267, 280, 329, 339, 400, 424, 449, 473, and it should be revised as “Figure x” to match the criteria of scientific reports.
  3. Some related literature can be cited in the category of “chemical sensors” on line 57 to help readers have a better understanding of the research background, like Analytical Methods, 2013, 5, 1612-1616 (DOI: 10.1039/c3ay26461k) and Organic Chemistry Frontiers, 2018, 5, 2170-3177 (DOI: 10.1039/c8qo00963e).
  4. The position of Figure 4 is recommended to be relocated to where line 249 is located, as Figure 4 was discussed by the sentences on line 249, and it is far away from where Figure 4 locates now near line 95.
  5. The sentence on lines 151-152 may need to be rewritten, as now it is not a complete one that reads “Biomedical technologies, environmentally responsive polymers are becoming valuable materials.”
  6. The letter “N” should be italic when used in a chemical name, such as on lines 153, 228, 229, 429.
  7. The resolution of Figure 8 is relatively low and makes it ambiguous. It is suggested to be replaced with a higher resolution figure.
  8. Figure 15 should be cited for the paragraph on lines 298-303 as it is the place where Figure 15 was discussed. Otherwise, Figure 15 was not mentioned in the whole manuscript.
  9. The Stern-Volmer constant of FPT shown on line 388 missed the multiple-sign, as it was written as “1.57 105 M-1”, which should be “1.57 × 105 M-1”.
  10. On line 435, the concentration of TPP should be 40 μg mL−1 with “-1” superscript. And on line 437, the λex and λem should have the “ex” and “em” subscript.
  11. The word “be” on line 448 should be deleted to make the sentence grammatically correct. On line 449, the semicolon after temperature should be deleted, and the period should be added after “(Figure 21)”.
  12. The words “Figure 22” on line 502 should be “Figure 24”, as this paragraph discussed the contents of Figure 24.
  13. An extra left parenthesis was found on line 521, which might be replaced with a comma.